# Key factors for effective implementation of healthcare workers support interventions after patient safety incidents in health organisations: a scoping review

Sofia Guerra-Paiva [1] , Maria João Lobão [1,2] Diogo Godinho Simões,[3,4] Joana Fernandes,[4] Helena Donato [5] Irene Carrillo [6,7] José Joaquín Mira [6,8] Paulo Sousa [1]

For numbered affiliations see end of article.

**Correspondence to**
Dr Sofia Guerra-Paiva;
sg.paiva@ensp.unl.pt

## ABSTRACT

**Objectives** This study aims to map and frame the main factors present in support interventions successfully implemented in health organisations in order to provide timely and adequate response to healthcare workers (HCWs) after patient safety incidents (PSIs).

**Design** Scoping review guided by the six-stage approach proposed by Arksey and O'Malley and by PRISMA-ScR.

**Data sources** CINAHL, Cochrane Library, Embase, Epistemonikos, PsycINFO, PubMed, SciELO Citation Index, Scopus, Web of Science Core Collection, reference lists of the eligible articles, websites and a consultation group.

**Eligibility criteria for selecting studies** Empirical studies (original articles) were prioritised. We used the Mixed Methods Appraisal Tool Version 2018 to conduct a quality assessment of the eligible studies.

**Data extraction and synthesis** A total of 9766 records were retrieved (last update in November 2022). We assessed 156 articles for eligibility in the full-text screening. Of these, 29 articles met the eligibility criteria. The articles were independently screened by two authors. In the case of disagreement, a third author was involved. The collected data were organised according to the Organisational factors, People, Environment, Recommendations from other Audies, Attributes of the support interventions. We used EndNote to import articles from the databases and Rayyan to support the screening of titles and abstracts.

**Results** The existence of an organisational culture based on principles of trust and non-judgement, multidisciplinary action, leadership engagement and strong dissemination of the support programmes' were crucial factors for their effective implementation. Training should be provided for peer supporters and leaders to facilitate the response to HCWs' needs. Regular communication among the implementation team, allocation of protected time, funding and continuous monitoring are useful elements to the sustainability of the programmes.

**Conclusion** HCWs' well-being depends on an adequate implementation of a complex group of interrelated factors to support them after PSIs.

### STRENGTHS AND LIMITATIONS OF THIS STUDY

⇒ The inclusion and exclusion criteria were defined in accordance with a preliminary search strategy, guided by the population, concept and context, as recommended by the Joanna Briggs Institute for scoping reviews.

⇒ We did not restrict language and period of time to avoid having selection bias and compromise the validity and reliability of the findings.

⇒ The data collection was limited to five interrelated dimensions (Organisational factors, People, Environment, Recommendations from other studies, Attributes of the support interventions).

⇒ We used the Mixed Methods Appraisal Tool assessment tool to evaluate the quality of the included studies; however, some of the criteria could not be fully applied in some specific cases.

⇒ We included five experts from different countries to complement the literature search with additional sources of information.

### INTRODUCTION

It is estimated that 10.4%–50% of the professionals working in healthcare sector will experience at least once in their career the second victim phenomenon (SVP)[1 2] defined as 'any healthcare worker (HCW), directly or indirectly involved in an unanticipated adverse patient event, unintentional healthcare error or patient injury, and who becomes victimised in the sense that they are also negatively impacted'.[3] These types of incidents, with an unintended or unexpected nature, can harm patients (first victims of an adverse event) or pose a risk to the system (near miss).[4 5]

HCWs play a crucial role in patient care and they can be seriously affected when a patient safety incident (PSI) happens. PSIs can impact HCWs' quality of life,[2 6 7] in

particular their physical and psychological well-being.[8][9] A study published in 2020 shows that the most prevalent symptoms in HCWs after PSIs were troubling memories, anxiety/concern and anger toward themselves.[9] Work satisfaction, confidence in their abilities[2] and work performance[7][10] can also be seriously impacted by these types of incidents. It can result in turnover intentions and absenteeism[11] and in the most severe cases can lead to suicide.[12]

Institutional support systems are increasingly being implemented in order to provide an immediate and empathic response to HCWs after stressful situations such as PSIs. Health organisations are recognising the importance of this type of support, due to its important impact on the organisational culture,[13] patient safety (PS) and quality of care[14-16] and also on the economic perspective.[17] It is well-established that poor HCWs' well-being has a strong influence on the reoccurrence of PSIs.[14] Therefore, prioritising interventions that effectively support HCWs after stressful situations can prevent future healthcare incidents and improve PS.

The first reported support programmes were implemented in the USA in 2006 and since then, they have been gradually multiplying all over the world.[18] In recent years, there has been a growing number of publications describing the implementation of these types of programmes and practices with the overall aim of decreasing emotional and psychological distress in HCWs. A systematic review found that HCWs seek support not only after being involved in PSIs, but also when facing other distressing situations (eg, emotional distress, torpid evolution of a patient, personal crises, intraoperative mishaps).[9] Based on the fact that there is still a lack of assistance to HCWs to cope in distressing situations, some support interventions are opening their scope of action.[9]

Although support interventions have demonstrated their benefits and utility, there is still limited research on finding what the common elements present in the development and implementation process of successful interventions are. A toolkit was introduced in 2010 to provide guidance on the implementation of programmes to support HCWs who have been negatively impacted by PSIs.[19] The development of this toolkit was an important step in assisting with the implementation of support programmes and it can be adjusted to any type of healthcare organisation.[19] However, no study has been published focusing on reviewing the existing evidence to understand the main factors that contribute for an effective implementation of these types of support interventions.

Evidence shows that establishing a set of elements for implementing interventions does not ensure its effective introduction into daily usage.[20][21] The success of interventions in health organisations highly depends on an adequate design, implementation and evaluation.[22] One of the main aims of implementation science is to understand what are the factors that might affect the effectiveness and sustainability of the interventions and what is the necessary implementation process to produce the expected effects.[20] In this sense, learning from previous

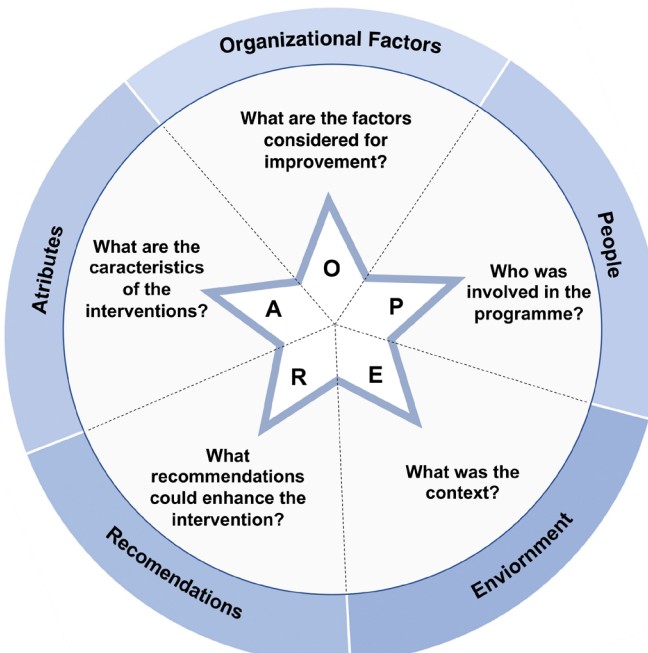

**Figure 1** OPERA—The five key domains to guide HCWs' support interventions after stressful events such as PSIs. HCW, healthcare worker; OPERA, Organisational factors, People, Environment, Recommendations from other studies, Attributes; PSI, patient safety incident.

experience can facilitate practical application and contribute to more effective interventions.[23]

### Study rationale

In this study, we set out to map and frame the main factors that underlie an effective implementation of support interventions in order to provide timely and adequate response to HCWs who are physically and/or emotionally affected by PSIs (known as second victims) or similar distressing situations. We have defined five interrelated dimensions guided by five main research questions, further described in this study, and we organised them in the Organisational factors, People, Environment, Recommendations from other studies, Attributes (OPERA) (figure 1). This framework helped to inform the planning and design of the scoping review, as well as the execution. The defined five domains were inspired on the health policy triangle (HPT) framework to guide effective implementation of health policies.[24] However, HPT is a theoretical model and in order to overcome the research-to-practice gap, we have incorporated the implementation science principles and Donabedian's structure–process–outcome quality of care model, more recently adapted by Yano.[25]

### Objectives

We aim to understand what existing organisational factors, relevant actors, contextual factors, operational attributes are present in interventions that were successfully implemented in health organisations to support HCWs after PSIs or other similar stressful events. We also

aim to identify what are the recommendations from the included interventions for improving the effectiveness of the programmes implementation in health organisations.

## METHODS

This scoping review is conducted using the six-stage approach proposed by Arksey and O'Malley[26] and is guided by the Preferred Reporting Items for Systematic Reviews and Meta-Analysis Extension for Scoping Reviews to ensure the transparency of the results obtained[27] and follows The Joanna Briggs Institute Methodology (JBI) for Scoping Reviews.[28]

All the methodological steps are described in further detail in the scoping review protocol published in a previous publication.[29]

### Stage 1: identifying the research question(s)

In this study, we focus on the main research question:

► What are the key factors that contribute to an effective implementation of interventions to support HCWs after PSIs or other similar stressful situations in health organisations?

To answer the primary research question, five secondary questions were formulated based on the specific objectives and outcomes of interest of the study :

► What are the organisational factors that contribute to an effective implementation of these interventions?
► Who are the relevant actors that contribute to an effective implementation of these interventions?
► What are the contextual factors that contribute to an effective implementation of these interventions?
► What recommendations, as identified in previous studies, can be applied to effectively implement these interventions?
► What are the operational attributes that contribute to an effective implementation of these interventions?

### Stage 2: search strategy

A comprehensive search strategy using relevant electronic databases was developed with the support of a qualified research librarian. The search comprised Medical Subject Headings terms along with free-text keywords. We applied the search strategy in nine electronic databases and the last update was done in November 2022 (CINAHL, Cochrane Library, Embase, Epistemonikos, PsycINFO, PubMed, SciELO Citation Index, Scopus, Web of Science Core Collection). The applied search strategies in the electronic databases can be consulted in online supplemental table 1. In addition to the database search, relevant websites were consulted and reference lists of the studies included in the full-text screening were screened to identify any other potential articles to include.

### Stage 3: study selection

We used EndNote to import articles from the different databases and we used Rayyan as a tool to facilitate the screening of titles and abstracts. The articles were

independently and manually screened by two authors between April 2022 and February 2023. In the case of disagreement on article inclusion, a third author was involved to evaluate the paper independently and contribute to making a final decision.

We did not restrict the period of time or language of the included studies in order to reduce the selection bias and to undertake a comprehensive overview of the existing literature on a topic with still limited number of publications. Empirical studies (original articles) were prioritised along with systematic reviews and meta-analyses for collecting potential eligible studies. Grey literature (theses and other documents) was also considered eligible for the study.

The inclusion and exclusion criteria were defined in accordance with a preliminary search strategy, guided by the population, concept and context (PCC) framework (recommended by the JBI for scoping reviews[28]) and are further described in the published protocol of this study.[29]

Based on the PCC framework, we defined the following criteria:

Population: Support interventions in health organisations in which HCWs are physically and/or emotionally affected by PSIs and other distressing situations. We considered support interventions destined to health professionals, residents and other allied health professionals (such as technicians and supply workers).

Concept: Support interventions that were fully implemented and executed in health organisations and provided measurable results that assessed the achievement of desired outcomes.

Context: Support interventions from a variety of healthcare contexts, including those in high-income, middle-income and low-income countries (eg, primary care, urgent and acute care, ambulatory services, long-term facilities).

### Exclusion criteria

Editorials, letters to the editor, case series, case reports, narrative reviews and commentaries were excluded.

### Stage 4: charting the data

A data extraction template was created to show the characteristics of the eligible studies (detailed information can be consulted in online supplemental table 2).

### Quality assessment

We used the Mixed Methods Appraisal Tool (MMAT) Version 2018 to conduct a quality assessment of the eligible studies.[30] We believe that this appraisal will be important to enhance the quality and rigour of our study, ensuring greater transparency and validity of the data. The eligible studies were evaluated by two independent reviewers. A third reviewer was involved in cases of disagreement in the quality assessment.

## Stage 5: collating, summarising and reporting the results

The information from the eligible studies was collected and organised into different conceptual categories, as presented in the OPERA (figure 1):

### Organisational factors
▶ Organisational structures (eg, infrastructures, resources, tools, equipment, units and staffing levels functional for managing and delivering services, leadership structure/authority and organisational culture).
▶ Organisational processes (eg, organisational actions, procedures, recruitment criteria, training, programme implementation, communication processes, quality of interactions and coordination during programme implementation and dissemination as well as the sustainability of the practice).
▶ Organisational outcomes (eg, implementation measures, process quality measures, utilisation measures, effectiveness measures that assess the attainment of an end state).

### People
Relevant actors (individuals and organisations that actively participate in the development and implementation of the programme).

### Environment
Contextual factors (type of healthcare setting and cultural context).

### Recommendations described in the included studies
Recommendations to improve the implementation process of the support interventions.

### Operational attributes of the interventions
Format/type of programme, accessibility, usability and confidentiality of the programme/interventio.

## Stage 6: consultation exercise and stakeholder involvement
We invited a group of five experts working on SVP research from five different countries (Finland, Germany, Italy, Portugal and Spain) to complement the literature search with additional sources of information. All of them are members of The European Researchers' Network Working on Second Victims (ERNST).

## Patient and public involvement and engagement
None.

## RESULTS
A total of 9708 records were retrieved from 9 electronic databases, 43 articles were retrieved from the reference lists of the included articles, 11 from websites and 4 were collected from stakeholders' group inputs.

Based on the screening of titles and abstracts, 7262 articles were excluded and 13 articles could not be retrieved

**Table 1** Characteristics of the included studies

| Categories | Subcategories | No of articles | Total |
|---|---|---|---|
| Type of scientific article* | Level II—evidence obtained from randomised controlled trial | 3 | 29 |
| | Level VI—evidence from a single descriptive or qualitative study | 26 | |
| Type of study design | Mixed method | 15 | 29 |
| | Quantitative descriptive | 8 | |
| | Qualitative | 3 | |
| | Randomised controlled trial | 3 | |
| Country where the study was developed | Denmark | 1 | 29 |
| | Germany | 3 | |
| | New Zealand | 1 | |
| | Spain | 2 | |
| | Sweden | 1 | |
| | UK | 2 | |
| | USA | 19 | |

*This rating scale is based on Ackley et al.[32]

after trying to contact the authors. A total of 156 articles were assessed for eligibility.

A third independent author was involved in solving four conflicts in the authors' decision, leading to the inclusion of one article. In total, 127 articles were excluded after the screening, and 29 articles ultimately met the eligibility criteria. Detailed information about the data collection, screening process, duplicates removed and reasons for exclusion is exhibited in the flow chart (online supplemental figure 1), in line with the original Preferred Reporting Items for Systematic Reviews and Meta-Analyses (PRISMA) statement.[31]

Studies with levels of evidence II and VI[32] met the eligibility criteria. We have included the following types of studies: mixed methods (n=15); quantitative descriptive (n=8); qualitative (n=3); randomised controlled trial (n=2) (for further details about the included studies consult online supplemental table 2). Bearing in mind that we only included empirical studies, we didn't include the first two screening questions in the MMAT evaluation (optional for MMAT).[30]

The characteristics of the included studies are outlined in table 1.

Most of the programmes included in this study have a multidisciplinary application and were focused on supporting HCWs after traumatic work experiences directly associated with PSIs. Several programmes were

**Table 2** Resources identified in the included HCWs' support programmes

| | |
|---|---|
| Marketing and dissemination materials | Print marketing materials: posters[40 59]; handouts such as brochures and flyers[39 45 55] identification badge of peer support for easy recognition and quick reference cards.[55] Digital marketing material: promotional videos[50] website[45 46 50]; email box.[41 46 47] |
| Selfcare and well-being related resources | Packets with aromatherapy[36 37]; chocolate[35 36 56]; snacks[51 54]; kind messages[35 36 51]; self-care pockets with essential support resources and guidance for coping with normal grief responses,[36] others contain a journal, a stress ball and tissues[56]; general mental and emotional wellness advices,[49] and use existing resources.[49] |
| Functional resources for programme implementation | Electronic mailbox[36 41–43 45–48 52 55]; access to virtual zoom[41 46] and WebEx platform[46]; dedicated mobile phone/pager/hotline/phone call system for peer supporter sessions[35 40 47 48 53 57 60]; web-based collaborative administrative platform for sharing information and managing the programme[45] such as Sharepoint[43 55]; checklist of responsibilities for the development team[36]; list of peer support schedules[38 39]; peer encounter forms[43 47] secure database of outreach attempts.[48] |
| Educational resources (most of them are related with peer support training) | Online training focused on psychological first aid[33 47] PowerPoint presentations with voice narrations[33] training scenarios[44 56 57]; videos[33 45 87]; 'Do's' and 'Don'ts' list, self-affirmations resources[48 87] and specific facilitator's guide[46] made available to peer supporters for guidance during the encounters with SV/HCWs; tutorial for peer support facilitation.[54] |

The most recent studies have an investment in administrative support resources such as the use of SharePoint, a collaborative platform for programme management.[43 45 55]
Posters, brochures and flyers were the most widely used marketing resources.
HCW, healthcare worker; SV, second victim.

particularly focused on responding to severe adverse events.[33–35]

Although one-to-one sessions were the most commonly provided support, some programmes also included group sessions. We also included interventions focused on raising awareness of SVP and creating a supportive and proactive culture to manage critical incidents and enhance HCWs' well-being.

The included interventions are described in online supplemental table 3.

In the following section, we present the results based on the organisation of the OPERA.

### (Organisational factors)PERA
### Organisational factors: structure
*Resources*
We found four main types of useful resources used in the interventions according to different applications.[36] These resources are described in table 2.

### Infrastructures
The acquisition of materials and human resources was, in most cases, voluntarily. However, some studies mentioned that the intervention received specific funding for acquiring resources.[36 37]

The existence of a specific room for sharing information and emotions in privacy was referred to by two studies.[35 38]

### Organisational culture
We identified the following factors associated with the organisational culture that are facilitators of the implementation of HCW support programmes after PSIs:
► Openness of the health organisation to innovation.[39]

► Implementation of previous initiatives that have contributed to the creation of a proactive organisational culture to manage PSIs[40], to support HCWs after PSIs /other stressful situations and promote their well-being.[36 37 41 42]
► The existence of formalised structures directed at fostering a PS culture, based on a just culture approach,[35 36 42] and at supporting HCWs and enhancing their well-being.[35 42–44]
► Active involvement of leadership members in initiatives that support SVs and HCWs' well-being.[35 37 39 42 45–48]
► Existence of established policies promoting a supportive organisational culture (such as the application of paid time off after a critical incident occurs)[35] and organisational accountability for employees' support and well-being after PSIs.[49]

We also identified some potential organisational barriers to the implementation of programmes:
► A lack of staff and leadership awareness regarding the support programmes for HCWs.[50]
► An organisational culture that does not prioritise PS and HCWs' support and doesn't disclose wellness problems.[51–53]

### Organisational factors: process
Most of the implemented programmes had developed a needs assessment[43 45 47 49 53 54] and/or conducted a literature review[33 45 48] prior to the design and creation of the programme. The needs assessment makes it possible to adapt the interventions to the needs of the clinical teams and adjust them to the institutional context and culture in accordance with the most recent literature.

The team was recruited using three different methods: direct nomination of the team members based on their ability to provide support in an empathic way[37 40 42 47 48]; votes from the clinical team[36 48] and voluntarily.[45 47 53]

An advertising campaign for raising attention of all the staff that would benefit the programme and show how to activate the service was carried out in a large number of interventions.[39 40 45 46 49 50 55]

We describe below some of the implemented communication strategies described in the included studies:

► Digital marketing: dissemination of the programme on computer screensavers[50] and digital communication through the institutional website.[39 46 50]
► Internal communication: hospital magazine, newsletters or email.[41 47 50]
► Networking: presentation of programme in divisional meetings[38 42 45 47 50 52] or in hospital-wide events and conferences.[38 56]
► Involvement of the leadership members in the dissemination process[50 51] and some programmes have included unit-level champions.[50]
► Previous staff training on the topic of SVP[45 50] including training for staff provides the first level of support after a PSI in the local.[57]

Dissemination was also carried out for recruiting peer supporters to join the peer support programmes.[42 43 45 48] Most of the peer supporters received specific training to prepare them for providing assistance to others.[36 37 39 40 42 43 45 47 48 50–52 54 55 57 58]

## Sustainability of the programmes

After implementing the pilot intervention, several projects have effectively expanded the pilot intervention to other departments,[43] other healthcare facilities[45 59] or hospital wide.[49 55] The full integration of the programme in the departments underlies the inclusion of the programme in the scheduled activities and in the available services of the institution.[57]

Leadership support was an important factor for the implementation of the programme and its sustainability.[40 47 50] Other programmes nominated unit champions to ensure the implementation of the programme and its sustainability by promoting a more supportive culture within the unit.[52 53 55]

Regular meetings were found to be important to maintain the cohesion of the team over time.[37 40 56 57] Annual courses and the implementation of an interactive virtual platform were important for the expansion of the workforce working in these programmes.[40 45] The high level of motivation and interest of the team[48] and retention of peer supporters were given particular consideration for programme sustainability and this was associated with work meaningfulness, staff satisfaction, commitment, a high level of resilience and a high level of confidence as a peer supporter.[50]

Some of the programmes were implemented in healthcare organisations where some PS initiatives and 'culture-shifting interventions' had already taken place.[37 39 50 55] In other cases they were integrated in major projects developed by the organisations.[42 56 60] Both situations were considered potential facilitators for maintaining the programmes over time.

Funding was also an important aspect to consider for the sustainability of several programmes.[37 42 46 52 59]

## Organisational factors: outcomes

Most of the studies included in the analysis focused on collecting outcomes related with programme's utilisation and the evaluation conducted by both peer supporters and users (HCWs/second victims that attended to the programmes).

In table 3, we describe in further detail the outcomes evaluated in the included studies.

## O(People)ERA: relevant actors

The establishment of a multidisciplinary team for the development and implementation of the support programmes was common to all the programmes. This team was predominantly composed by leadership members (hospital administrators and unit leaders), front-line workers, academics and experts in quality and safety.[33 34 38 43 50 52] In some cases, it also included chaplains,[38] social workers[38 43] and legal department members.[52]

Most of the programmes' development and implementation were dependent on volunteer efforts. However, some programmes hired specific elements of the team, such as the programme directors and coordination members.[39 42 57 60] Several studies highlighted the importance of these members in the programme activation process, particularly in matching the profile of peer supporters with HCWs' needs and in the contact with peer supporters and outreach people in need of support.[42 48 52 53 56] One programme included contract freelancer work by psychotherapists to provide more specialised support[60] and other programmes remunerated peer supporters for their work.[37 49 60]

Trained peer supporters are crucial for providing effective support for HCWs involved in PSIs. Specialised trainers from different types of backgrounds (such as psychology, nursing, quality improvement and PS, workplace wellness, legal services, executive sponsors, department representatives) provided workshops and seminars for peer supporters.[36 43 48] In some cases, there was a specialist to facilitate monthly debriefing meetings for peer supporters to process their experiences and to receive assistance.[36 37 43]

Most of the programmes also provided access to specialised external support that represents the third level of support in the case of programmes that follow the Scott Three-Tier Model.[34 36 45 51 57] In other types of programmes, complementary support was provided by chaplains, social workers or Employee Assistance Programme counsellors.[35 38 43 45]

On a department level, unit leaders performed different types of essential functions by contributing to

**Table 3**  Collected outcomes from the included interventions

| | |
|---|---|
| Outcomes related to support services utilisation | Frequency of the HCWs who attended the programme[36 40 42 60 87]; frequency of programme activation[34 37 40 42 43 45 47–49 51 52 55–57]; average duration of the encounters[57]; no of programme dropouts[33]; median no of interactions per month[52 53 57]; frequency of peer support encounters[50 53]; no of HCWs who need external support.[57] |
| Evaluation of the programme by the peer supporters perspective | Overall peer support satisfaction with the training[33 34 47 49 87]; perception of acquired knowledge, meaningfulness, motivation and interest to learn more and apply the learning[44]; satisfaction about how encounter end out[45]; need for additional training and experience[45]; feeling able to provide support and being comfortable with their knowledge and skills as a peer supporter.[45] |
| Evaluation of the programme by the user perspective (HCWs involved in PSIs/SV) | Overall satisfaction with the programme[33 36–38 40 43 45 49 51 54 58 59 87]; knowledge/skills acquisition[33 41]; usefulness of the contents[33]; timeliness of the programme[48]; perceived helpfulness of the programme[46 53]; HCWs awareness of SVP phenomenon[39]; qualitative experience after attending the programme (how much HCWs benefit from the programme).[33 36–38 40 43 45 49 51 54 58 59 87] |
| Health-related outcomes | Psychological and physical distress[47]; emotional distress[46]; perceived stress[44 60] anxiety and burn-out[54 58]; assessment of quality of life[36]; perceptions of individual coping skills such us emotion regulation, self-efficacy and resilience.[39 44 47 87] |
| Work-related outcomes | Job satisfaction[36]; turnover intention and absenteeism[47]; return to work[35]; confidence in coping with adverse events.[87] |

HCWs, healthcare workers; PSIs, patient safety incidents; SVP, second victim phenomenon.

the development of the programme,[37 48] participating in the recruitment of peer supporters,[33 43 50 51] providing first-level support for HCWs in need,[34] coordinating programme's components and mentoring peer support team members within the facility.[57]

### OP(Environment)RA: contextual factors
On an internal level, we found that most of the programmes were implemented in large and academic hospitals, characterised by an environment with multiple and complex divisions,[34 35 39 40 42 45–47 52 55 57] and with a high level of specialisation (tertiary and quaternary care).[37 41 48 49 51] Some programmes were specifically implemented in stressful and busy environments, such as emergency medicine departments, intensive care units and psychiatric departments.[39 44 51 54 55]

We found that in some cases the organisational environment was beneficial to the implementation of the programme, particularly when healthcare organisations were already working towards creating a more supportive environment for their staff and strengthening the safety culture.[36 37 42 43 56 60]

We also found that previous occurrence of a very serious adverse event helped in recognising the need to implement a programme to support staff in supporting them to cope after PSIs.[35 50]

On an external level, several studies have mentioned that programme implementation was affected by the COVID-19 pandemic, such as the possibility of handling face-to-face encounters, and it also affected the data collection/monitoring process of the interventions.[6 8 9 11 22 24 27]

### OPE(Recommendations)A: recommendations related with the implementation process and directly related with HCWs experience
Different types of recommendations for improving the programmes were mentioned in the included studies with the ultimate goal of achieving a more effective intervention. They were identified from the user perspective (HCWs in need of support after being involved in a PSI) and also from the perspective of the implementation process (described in table 4).

### OPER(Attributes): operational attributes of the programmes
#### Accessibility
The access of HCWs to the programmes was done by different channels to find the most convenient format for the users: phone call,[40 43 50 55 60] email[41 43 44 55] or direct contact with peer supporters or with core team members.[43 44 55]

Programmes can be activated by the following people:
▶ Anyone who was involved in the stressful event.[34 42 47 49 56 57]
▶ Safety and risk management staff.[52 56]
▶ Peer supporters.[52]
▶ Nurse in charge.[38]
▶ Leadership members.[34 35]
▶ Programme directors.[42 48 52]

In some cases, the entire clinical team is contacted by the implementation team or by leadership members with a view to integrating in the support programme after a PSI, however, acceptance only depends on the individual choice of the HCWs.[34 48 53 58]

In the case of programmes that have online resources, they could be accessed through a website.[33 39 45]

Although most of the programmes were provided voluntarily, some of them have mandatory activities for

**Table 4** Main recommendations referred to in the included studies from both the user and the implementation process perspectives

| Recommendations related with the implementation process | Recommendations directly related with users' experience (HCWs involved in PSIs/SV) |
|---|---|
| Conditions to facilitate the implementation process:<br>To allocate protected time for teams to implement the programme and actively participate in the tasks and training.[34 45]<br>Administrative framework should be ensured to support programme implementation.[57]<br>To develop an institutional policy to guide the management of the critical event and support the affected HCWs and patients.[35]<br>Funding was an important facilitator for programme development and implementation.[37 46 55]<br>To be formally recognised as an institutional programme.[34]<br>To invest in telehealth solutions to support HCWs in the workplace.[46]<br>Procedures related with the implementation process:<br>To invest in programme's dissemination and marketing for increasing HCWs' adherence to the programme.[34 46–49 51 55 57 60]<br>To actively involve the target group in the development of the programme and to conduct a needs assessment helps fostering interest and adapt to the specific needs of the target population.[54 58]<br>To integrate staff working in the unit in the programme's team, helps to understand the needs of the unit.[37]<br>To promote active involvement of leadership members facilitates the implementation of the programmes and contributes for staff engagement.[33 35 40 42 45 49 51 54 56 57]<br>To promote training sessions and resources to increase managers awareness about the SVP and about the existing support programmes.[51 56]<br>To create a multidisciplinary support team to facilitate a comprehensive programme's development and address different areas for support while leveraging a range of expertise.[52]<br>To set regular debriefings (in person or virtual meetings) to exchange experiences and to foster a culture of mutual support among the members of the programme's team.[40 56]<br>Training should be provided to peer support training and role play is one of the most recommended formats.[34 50 56]<br>To develop a list of key phrases that peer supporters can use in their interactions with SVs.[50]<br>To evaluate the impact of the programme and monitor its longer term effects and drive continuous improvement.[37 43 44 46 47 49 50 56]<br>To use pre-existing structures, resources and adapt existing programmes to facilitate the development and implementation of the programme.[45 49 55 56]<br>Having an electronic dashboard for sharing documentation and data collection.[55] | Conditions to facilitate users' experience:<br>To allocate protected time for HCWs to attend the programme's activities and the support sessions.[54 59 87]<br>Participation in the process should be entirely voluntary and confidential.[34 40 45 52–54 57 87]<br>To invest in creating an organisational culture that addresses and acknowledges clinicians' vulnerability, while promoting a supportive environment after stressful incidents.[42 57]<br>To involve legal and risk management departments to address concerns about confidentiality or related issues.[34 35 52]<br>To ensure that SV have an adequate access to the programme, feel safe and not stigmatised when accessing resources[51];<br>Procedures focused on user's experience:<br>Appropriate timing for programme's activation (ideally it should be immediately available to the HCWs after a stressful event[57]) and adequate duration of the support.[40 87]<br>All HCWs involved in a critical incident should be contacted to receive support.[40 52]<br>Active surveillance in the units should be done to identify potential SV (particularly in high risk environments).[57]<br>To increase programme awareness for front-line staff and prepare them how to give first level of support.[43 47 51]<br>To train the leadership to support staff to cope with stressful situations and to direct them to support resources in case of need.[34 36 37 56 87]<br>To enhance the level of education on staff resiliency in the services.[39]<br>To establish an interdisciplinary support team to open the scope of support according to the different staff's needs and background.[43 50 55]<br>To create a safe place for sessions.[36–38 54]<br>Smaller groups are preferred for sharing experiences and support.[59]<br>To make resources available and close to the staff, to make them easy to reach.[38 42 51]<br>To identify barriers and facilitators for HCW to seek mental health support (eg, stigma, career concerns, protected time).[46]<br>To provide channels to reach the programme even when the HCW was not involved in a PSI.[48]<br>To provide a holistic support (eg, incorporating integrative therapy techniques; cognitive based therapy).[37 44] |

HCWs, healthcare workers; PSIs, patient safety incidents; SV, second victim.

all staff in the departments, such as a seminar to promote a shared understanding of SVP and the need for peer support,[49] or attendance at debriefing sessions to enhance the recovery of all staff in the unit.[35]

Programmes based on the Scott Three-Tier Model establish the access to the programme according to different levels of HCW needs.[34 40 57] The first level of support should be available immediately after the incident has happened and team members should be prepared to provide it (local-level support). For accessing the second level of support, anyone can activate peer support with a trained peer. For the third level of support, the HCW is referred to specialised support.[57]

In some cases, programmes can be accessed 7 days a week.[39 43 46 47 55 57 60] In a support programme implemented in New Zealand, the phone number to reach the support team was added into the staff contact list and on-call phones within the unit to facilitate the activation process.[40] The dissemination of schedules, timelines and contacts was a useful strategy employed to facilitate access to the programme.[39 47]

Lack of staff awareness about the programme[50 51] and difficulty finding time to attend the interventions were the main barriers to accessing the programmes.[59] Moreover, some HCWs resisted accessing support since they did not recognise the need for it or preferred to avoid dealing with the situation again by talking about it.[35]

## Usability

To ensure the maximal attendance of staff, in several programmes the communication process was facilitated by peer supporters or the programme director

after the activation of the programme. These actors have the responsibility of contacting the HCWs immediately after the incident to schedule a one-to-one peer support encounter.[42 45 48 49 52 53] In several programmes, the schedule was negotiated with HCWs according to their needs,[37 40 45 48 52 53 56] and in some cases the location[37 45] and format (in person, phone call or email) of the sessions were also negotiated.[48 56]

In some cases, staffing relief at the workplace and protected time were provided to allow HCWs to attend the support programmes during their working time.[37 38 41 45 54 58]

To facilitate initiation of the programme, a leadership member or the programme director matched the peer supporter profile with the HCWs' needs according to the different criteria, such as geographic proximity and the same medical specialty.[45 48 56] In other situations, such as in the Buddy Study Programme, the HCWs selected the peer supporter according to their preferences so they could contact if they needed support.[49]

To facilitate the HCWs' participation in the programme, some programmes prioritised the virtual format[41 45 46 52] and limited the duration of sessions to 60 min.[37 40 46 54 59]

A peer support intervention in Sweden defined the topics to approach in the sessions based on participants' suggestions and beliefs.[58] This made it possible to attend to the needs of the participants and to promote their adherence to the interventions.[58]

In online programmes such as MISE (Mitigating Impact in Second Victims, we found that browsing, amount of time required to complete the activities and comprehension of the programme content were valued attributes to facilitate the usability of the programme.[33]

## DISCUSSION

In this study, we focused on identifying the highest number of reported programmes and other initiatives to support HCWs after PSIs. The majority of the included programmes are based on peer support. These types of programmes are founded on social support with the ultimate goals of increasing professional quality of life, decreasing emotional stress and ensuring patients' safety.[61–63]

We found that these types of programmes should be voluntary, with easy access and widely disseminated in healthcare organisations. This will make it possible to provide immediate psychological first aid after a distressful event and to overcome obstacles related to a lack of awareness of SVP and stigmatisation associated with HCW vulnerability.[50 55 63–65]

The association between just culture and empathic and non-judgemental responses to PSIs has been clear.[18 50 66 67] This is identified as a core condition for an open communication, and to establish positive relationships between peer supporters and HCWs, thereby enhancing programme adherence.[67 68] Namely, willingness to give support with empathy, leadership skills, reliability, being

communicative and not being judgemental are essential elements for effective peer support.[62] Another study pointed out that the ability to understand others' feelings and experiences after a PSI can be beneficial to the support process, by improving emotional regulation and reinforcing the cooperation between HCWs and peer supporters.[69] In our study we have identified that all these principles were mentioned to improve the effectiveness of the interventions and, therefore it should be taken in consideration in peer supporters training when implementing the programme.

We found that specific training for peer supporters was provided in most of the programmes to prepare them to adequately provide psychological support according to HCWs' needs. Training is believed to be one of the key components to consider when implementing a support programme.[50 55 63] According to implementation research, it is essential for an effective programme implementation.[70]

Many of the included studies also mentioned the importance of setting regular multidisciplinary meetings to share important learning and experiences and to keep the team motivated over time. Rosak-Szyrocka points out that having a motivated team is very important to ensure their commitment to and engagement in work.[71] This study also points out that in hospitals, HCWs are particularly motivated by strong interpersonal relations and a positive atmosphere, as they foster cooperation and mutual support among the team members.[72] Another study indicates that multidisciplinary teamwork is an essential element for improving outcomes at an organisational level.[73]

We found that active participation by leaders in the initiatives can influence the effectiveness of programmes implementation and their sustainability. Helping to create a safe and resilient environment can increase the programmes' acceptability among HCWs and their engagement in the activities.[73–76] Leaders' participation was not only important in the implementation process, but also in the development of the programmes. In particular, it can contribute to adjust the programmes to the healthcare context, facilitate the acquisition of resources and recruitment of peer supporters.

We also found that leadership engagement in HCWs/SVs support initiatives is very important for strengthening the organisational culture towards a non-punitive response to error. Leadership members are essential actors in helping to create stimulating and supportive environments in healthcare among teams.[72 77] Boguslavsky et al[74] refer the importance of having leaders with empathic and communication skills, that are able to listen, empower and encourage others, in line with non-blame culture principles. Therefore, the involvement of leaders with these types of skills will benefit the support programme's implementation.

Moreover, we also found that the success of the implementation is also dependent on how closely aligned it is with HCWs' needs. Therefore, HCWs should be consulted

and involved in the programme development process. Our results corroborate Søvold *et al*'s findings, which highlight the importance of HCWs participating more in the decision-making process as well as in the development, implementation, testing and evaluation of the interventions, with the ultimate aim of improving their health, well-being and job satisfaction.[78] Interventions that include the target population's perspectives from the first steps of development exhibit a higher level of adherence and adoption and are more sustainable over time.[70]

However, HCWs are very often overloaded with duties that could be undertaken by other staff, and prevented from performing other necessary tasks that need their qualifications, with few opportunities to apply for training, develop their professionals skills and be available to participate in workplace initiatives.[73] This contributes to less work satisfaction and more costs for the system.[73] We found that working conditions are one of the main pillars for ensuring effective implementation and its sustainability. Providing protected time and staffing relief were identified as two of the main priorities for ensuring that, on the one hand, qualified HCWs could participate in the development and implementation of support interventions, and on the other, HCWs would be able to attend the support programmes if they needed support. Financial incentives play also an important role in keeping the support teams over time and to facilitate the programme implementation. Some evidence corroborates these findings, namely in what concerns to HCWs retention.[70 73]

Finally, it is agreed that monitoring programme outcomes over time is essential for assessing programme effectiveness, and evaluating its progress and impact on HCWs/SVs and in health organisations.[79] This process should be continuous and facilitates the ongoing improvement of the programmes.[22] However, there are a limited number of studies that follow-up interventions over time. According to Wade *et al*, it is still not clear how much time would be necessary to monitor the impact of programmes on HCWs' skills and knowledge.[80] Thus, we recommend that future programmes invest in monitoring their results over time and for longer periods.

In hospital settings, we have found that programmes frequently monitor before and after interventions, by collecting both qualitative and quantitative data. It is agreed that monitoring process should follow rigorous and feasible when assessing both types of measures.[79]

Although hospital settings are increasingly investing in the monitoring process, the study of the impact of HCWs' support programmes it is still limited and unreported in non-hospital settings.[81] This might be due to the insufficient emphasis on safety culture in non-hospital settings,[81 82] which could be improved by increasing PS initiatives and awareness campaigns in these contexts.

The experience from support programmes directly and not directly associated with supporting HCWs after PSIs was particularly useful to identify barriers and challenges in the access and adherence to the programmes and sustainability concerns. One of the main topics of discussion is focused on programmes' confidentiality.[50 83–85] Wade *et al* highlighted the need to establish a consensus and build a body of evidence to evaluate these types of programmes in ethically and confidentially which involves protecting the privacy and confidentiality of the attendees of these types of programmes.[80] We also highlight that the legal framework of each country can influence the success of the programmes' implementation. Professional liability is often not cited in articles describing interventions, despite its influence on the programmes and its impact on transitioning from a reactive safety culture to a generative safety culture.[86]

Ultimately, we found that these types of programmes should be formalised and have defined structures to facilitate its sustainability and to overcome potential institutional barriers to the implementation of the programmes. Examples, such as RISE (Resilience in Stressful Events) programme, have demonstrated that the formal recognition of the support programme in all the large academic medical centre, the use of existing structures and involvement institutional stakeholders have not only strengthen its visibility within the institution, but also inspired other external health settings to implement their own support programmes.[50]

In summary, we have organised the main findings in online supplemental table 4, organised according to the OPERA.

## LIMITATIONS

In this study, we have found that long-term evaluation of programmes it is still limited for the most part, and in some cases, evaluating the outcomes is not recognised as a priority for reasons of confidentiality and ethics. We recognise the need to strengthen the organisational culture towards a non-punitive response to error in order to overcome potential barriers to programme adherence and evaluation. It is very important to increase the follow-up time to understand the impact of these types of programmes and their effectiveness in the long term.

We evaluate all the included studies using the MMAT quality assessment tool since this tool is suitable for different types of methodologies, however, some of the criteria could not be applied in the descriptive studies. We suggest that this tool could be adapted in the future to these types of studies.

## CONCLUSION

This is the first time that a study has focused on understanding the set of characteristics and elements necessary for a successful programme' implementation to respond to HCWs needs after PSIs, based on the fact that their success highly depends on an adequate implementation and evaluation process.

This study was inspired in the previous experience from other support programmes with the ultimate propose of guiding the implementation of HCWs support

programmes in health organisations and contributing for future evidence-based practice.

In summary, we concluded that programmes should be easily accessible and voluntary for all HCWs in health organisations. Dissemination should be prioritised in order to give higher level of visibility to these programmes. The effectiveness of programmes' implementation is highly dependent on the organisational culture, the active involvement of leadership and a multidisciplinary team. Training should be provided for both peer supporters and leadership members, to make it possible to respond to HCWs' needs in a more prepared and satisfactory way. Regular communication among support teams should be maintained over time to keep teams motivated and increase their retention. Moreover, it is recommended to allocate dedicated time and staffing resources to engage in these types of interventions. Establishing formalised structures and securing funding sources it is important for the programmes's sustainability. The use of existing resources can overcome potential institutional barriers. Finally, programmes should be monitored for their continuous improvement without compromising the confidentially of the data.

### Author affiliations
[1]Public Health Research Centre, Comprehensive Health Research Center, CHRC, NOVA National School of Public Health, NOVA University Lisbon, Lisboa, Portugal
[2]Internal Medicine Department, Hospital de Cascais Dr Jose de Almeida, Alcabideche, Portugal
[3]Public Health Unit of ACES Almada-Seixal, Almada, Portugal
[4]NOVA National School of Public Health, NOVA University of Lisbon, Lisbon, Portugal
[5]Documentation and Scientific Information Service, Centro Hospitalar e Universitário de Coimbra, Coimbra, Portugal
[6]Health Psychology, Miguel Hernandez University, Elche, Spain
[7]Health Psychology, FISABIO, Miguel Hernandez University, Elche, Spain
[8]Salud Alicante-Sant Joan Health District, Elche, Spain

**Acknowledgements** This work was supported by ERNST–Cost Action 19113 (COST–European Cooperation in Science and Technology, www.cost.eu). We would like to particularly thank the members of stakeholder groups, namely Prof. Reinhard Strametz, Dr Susanna Tella and Professor Massimiliano Panella, for their contribution to this study as members of the consultation group.

**Contributors** SG-P was involved in the writing of the article and is responsible for the overall content as guarantor. SG-P, MJL, PS and JJM contributed to the conceptualisation of the study. SG-P, MJL and DGS participated in the pilot testing stage and defined the inclusion criteria. SG-P, MJL, DGS, JF and IC were involved in the screening process of the eligible studies. HD contributed to the definition of a robust search strategy and participated in the preliminary search in relevant bibliographic databases, particularly in peer-reviewed journals and grey literature databases. PS, JJM and IC contributed to the literature review. PS and JJM also participated in the consultation group. All authors read and approved the manuscript.

**Funding** The present publication was funded by Fundação Ciência e Tecnologia, IP national support through CHRC (UIDP/04923/2020) and under the grant (UI/BD/150875/2021).

**Competing interests** This protocol is part of the PhD project of the corresponding author (SG-P), which has been previously approved by the National School of Public Health (ENSP-NOVA). Some of the research group members are part of the ERNST-COST Action 19113. All the authors have submitted the disclosure of interest form.

**Patient and public involvement** Patients and/or the public were not involved in the design, or conduct, or reporting, or dissemination plans of this research.

**Patient consent for publication** Not applicable.

**Provenance and peer review** Not commissioned; externally peer reviewed.

**Data availability statement** All data relevant to the study are included in the article or uploaded as online supplemental information.

**ORCID iDs**
Sofia Guerra-Paiva http://orcid.org/0000-0002-4311-3799
Maria João Lobão http://orcid.org/0000-0002-3837-4476
Helena Donato http://orcid.org/0000-0002-1905-1268
Irene Carrillo http://orcid.org/0000-0002-6981-7284
José Joaquín Mira http://orcid.org/0000-0001-6497-083X
Paulo Sousa http://orcid.org/0000-0001-9502-6075

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
