## [Reviewer comments · BMJ Open]

ARTICLE DETAILS

TITLE (PROVISIONAL)	Key factors for effective implementation of healthcare workers support interventions after patient safety incidents in health organisations - a scoping review
AUTHORS	Guerra-Paiva, Sofia; Lobão, Maria João; Simões, Diogo; Fernandes, Joana; Donato, Helena; Carrillo, Irene; Mira, José; Sousa, Paulo

VERSION 1 – REVIEW

REVIEWER	Selamu, M Addis Ababa University, Psychiatry
REVIEW RETURNED	09-Aug-2023

GENERAL COMMENTS	This review is well written and good to be published but requires clarification on a. who are health care workers are they nurses, physicians operationalization is needed b. the context of the health system also requires some clarification c. clarification on the patient safety incident is also useful what does patient safety incident mean?
---

REVIEWER	Irwan, Andi Hasanuddin University
REVIEW RETURNED	02-Sep-2023

GENERAL COMMENTS	Title: please shorten the title into Population, Concept, Context (PCC) only Introduction: please mention the limitations of previous review articles to show the originality of this current review Methods: - please mention explicitly what is the PCC in this review?- What are the inclusion and exclusion criteria of this review?- What is the reason for "In this study we didn't restrict the period of time or language of the included studies" ? - At the study selection, did you use specific software or did you do it manually? How many articles are in disagreement during study selection? please explain the process in detail to ensure the interrealibility of the review. You may refer to the JBI manual.
--

REVIEWER	Roche, Dominic Cardiff University
REVIEW RETURNED	19-Sep-2023

GENERAL COMMENTS	Thank you for submitting this paper for peer review. This is an important topic and you have presented some interesting and relevant evidence in your scoping review. Please can I refer you to
---

the following points to consider to and which I hope you find helpful in further develop this manuscript.

1. Is the research question or study objective clearly defined?
No.

You set out 'objectives' on page 5 – this section isn't quite congruent with the main goal and objectives outlined on page 6 'identifying the research question' (for example the statement relating to collecting recommendations from previous studies). I would also like to read more about the interrelated dimensions you refer to in this 'objectives' section on page 5 and how this helped to inform the planning, design and execution of your scoping review.

Can you also clarify if the aim of the review is to examine factors / issues relating just to the implementation of the programme / interventions, or are you also interested in outcomes and if so, what are these outcomes beyond implementation? In regard to this, I note that the conclusion presented in your abstract relates to HCWs wellbeing which does suggest you are interested in more than implementation factors.

I would question the use of the term 'key factors' in the main aim – perhaps 'factors' more broadly? Please be consistent with use of 'aims', 'goals' and 'objectives' as these appear to be used interchangeably. In relation to the secondary 'objectives' – can you clarify why you are only focusing on interventions relating to when HCWs are physically and / or emotionally affected by PSIs? What is the trigger for the intervention(s) to recognise this has occurred? I think more background and contextual information in the introduction would more effectively position this scoping review in the context of the stated aim and objectives. This would include more detail about the second victim phenomenon in the context of patient safety incidents and the potential impact on healthcare workers. I'd also like to read more about the theoretical underpinnings of the interventions / programmes you refer to in your introduction. How have these come about – what is the proposed 'mechanism of action' to support healthcare workers in these situations? For example, you note that there are programmes / interventions designed for decreasing emotional and psychological distress in HCWs – what specifically issues / context are particularly relevant when considering PSIs?

You focused on five key domains using the acronym OPERA, guided by the five main questions – are you able to provide more detail here as to how you came to decide on these key domains? I also note on page six that you have included "support interventions in health organisations in which HCWs are physically and/or emotionally affected by PSIs and other distressing situations based on the fact that most support interventions end up being directed toward these types of programmes." I think this statement needs to be developed in more detail with supporting evidence and perhaps examples of what exactly you mean here and how this relates to the aims and outcomes of your review.

2. Is the abstract accurate, balanced and complete?

Yes. With the caveat that you refer to 'effective implementation', whereas your review appears to have addressed issues beyond implementation (or there needs to be more clarity regarding how you have conceptualised implementation in the context of this review). The aim could be separated from the background section so it is more clearly identified.

	3. Is the study design appropriate to answer the research question? Yes, a scoping review is appropriate to address the aim and objectives. 4. Are the methods described sufficiently to allow the study to be repeated? No. However, I note the study protocol has been reported in a separate publication providing additional level of detail. 5. Are research ethics (e.g. participant consent, ethics approval) addressed appropriately? N/A. Ethical approval not required, but you could refer to ethical conduct / ethical issues more broadly in the context of the overall study and the included papers. 6. Are the outcomes clearly defined? No. As noted in my response to question one, I found it a little challenging to identify what the specific outcomes of interest were. You have stated you are interested in factors related to the implementation of these programmes, but this doesn't appear to be congruent with some of the objectives stated on page 6 as you then refer to factors contributing to effective support programmes. Do you mean effectiveness of implementation or effectiveness of programme outcomes (or both)? 7. If statistics are used are they appropriate and described fully? N/A 8. Are the references up-to-date and appropriate? Yes. Some referencing errors (Ward). 9. Do the results address the research question or objective? Yes. 10. Are they presented clearly? Yes. However, you have used the acronym OPERA to guide your results and provide a helpful figure in support of this You could also more clearly sequence your results to reflect this acronym, which would help provide a more logical structure to the results section. I would also encourage you to make more effective use of numbering and / or sub-headings in your results section to help guide the reader through the respective sections. (NB on page 7, 'stage 5' you could also present the main categories in the 'OPERA' sequence to provide a more logical structure and clearly identify 'attributes'.) The results are in some cases presented using a list / bullet point approach. I think it would enhance the readability of your results section if you consistently adopted a narrative approach to the presentation by synthesising the points of discussion together to help more effectively represent the respective themes and sub-themes. It's unclear to me why you have chosen to present the 'recommendations' theme as a table, whereas the other themes are not presented in this way. (I wonder if this is related to word count limits?) I did find this approach disrupted the flow of your results – are you able to report the recommendations as a narrative, perhaps supported by supplementary information in a table?
--	--

	11. Are the discussion and conclusions justified by the results Yes. However, there are some short paragraphs which disrupt the flow of the discussion section – this is particularly evidence toward the latter stage of the section. In regard to the conclusions, I wonder if you would be able to develop a model / framework that can demonstrate to the reader what key factors are required for effective implementation after PSI's in health organisations? This would then clearly address the originally posited question for your review. I think you could also develop in more detail the issue of non-hospital based interventions – what could be done to address the lack of evidence in this regard? I also think it would be helpful to consider more overtly during your discussion and concluding remarks what we already know about other support mechanisms and interventions for staff that are not specifically related to PSI and the SV phenomenon and what learning we could take from this evidence base. 12. Are the study limitations discussed adequately? Yes. 13. Is the supplementary reporting complete (e.g. trial registration; funding details; CONSORT, STROBE or PRISMA checklist)? Yes. 14. To the best of your knowledge is the paper free from concerns over publication ethics (e.g. plagiarism, redundant publication, undeclared conflicts of interest)? Yes. 15. Is the standard of written English acceptable for publication? Yes.
--	---

VERSION 1 – AUTHOR RESPONSE

Reviewer: 1

Dr. M Selamu, Addis Ababa University

Answer from the authors to a) and b):

Dear Dr. M Selamu, thank you for your comments.

Following your suggestion we have included the requested information on the study selection section (lines 185-189, page 6):

“We considered support interventions destined to health professionals, residents, and other allied health professionals (such as technicians and supply workers) in a variety of healthcare settings, including urgent and acute care , primary care, ambulatory services, rehabilitation centres, long-term facilities, and outpatient surgery centres. The included support interventions were fully implemented and executed in health organisations and provided measurable results that assessed the attainment of a desired outcome.”

In order to respect as much possible the recommend length of the article according with BMJ guidelines, we didn't include most of the aspects that are described in the published protocol of the study (<https://bmjopen.bmj.com/content/12/8/e061543>). In this sense we have described in the lines 177-179 (page 6): the following information :

"The inclusion and exclusion criteria were defined in accordance with a preliminary search strategy, guided by the population, concept and context (PCC) (recommended by the JBI for scoping reviews (21)) and are further described in the published protocol of this study (20)."

Answer from the authors to c):

Dear Dr. M Selamu, thank you for pointing out this need for improvement. The definition of a patient safety incident is outlined in the protocol. However we've incorporated it into the manuscript based on your recommendation (lines 79,81 of the manuscript- 1st paragraph of introduction section).

Reviewer: 2

Dr. Andi Irwan, Hasanuddin University

Comments from the Reviewer:

Title: please shorten the title into Population, Concept, Context (PCC) only

Answer from the authors:

Dear Dr. Andi Irwan, thank you for your review and comments.

In this scoping review, our population is "healthcare workers support programmes after patient safety incidents". For concept we have considered "key factors for effective implementation" and for context "health organisations".

Our title was finally organized in "Key factors for effective implementation of healthcare workers support interventions after patient safety incidents in health organisations -a scoping review". We would really appreciate if the title will be the same as the protocol of the study also published in BMJ Open in order to be related with the previous publication.

Comments from the Reviewer:

- Introduction: please mention the limitations of previous review articles to show the originality of this current review

Answer from the authors:

Dear Dr. Andi Irwan,

Thank you for your comment. Based on your feedback we have described the limitations from previous studies and the originality of this study in the 5th paragraph of the introduction section:

Lines 102-104 (page 4) - "Although support interventions have demonstrated their benefits and utility, there is still limited research on finding what the common elements present in the development and implementation process of successful interventions are."

Lines 107-109 (page 4) - "However, no study has been published focused on reviewing the existing evidence to understand the main factors that contribute for an effective implementation of these types of support interventions and its success."

Comments from the Reviewer:

Methods:

- please mention explicitly what is the PCC in this review

Answer from the authors :

Dear Dr. Andi Irwan,

Thank you for your comment. In order to respect as much possible the recommend length of the article according with BMJ guidelines, we have reported the population, concept and context (PCC) in more detail in the published protocol of the study (<https://bmjopen.bmj.com/content/12/8/e061543>).

Comments from the Reviewer:

What are the inclusion and exclusion criteria of this review?-

Answer from the authors :

Dr. Andi Irwan ,

As recommended we have increased the level of detailed in the description of the inclusion criteria:

"In this study we focused on identifying the highest number of programmes and other initiatives focused on supporting HCWs after PSI's. Therefore we included support interventions in health organisations in which HCWs are physically and/or emotionally affected by PSIs and other distressing situations. Based on the fact that there is still a lack of assistance to HCWs to cope in distressing situations, some support interventions are opening their scope of action (9)." (Lines 180-184, page 6)

"We considered support interventions destined to health professionals, residents, and other allied health professionals (such as technicians and supply workers) in a variety of healthcare settings, including urgent and acute care , primary care, ambulatory services, rehabilitation centres, long-term facilities, and outpatient surgery centres. The included support interventions were fully implemented and executed in health organisations and provided measurable results that assessed the attainment of a desired outcome." (Lines 185-189, page 6)

This information has been also reported in the study protocol in a separate publication providing additional level of detail.

We have described the following information in the manuscript (lines 177-179, page 6):

"The inclusion and exclusion criteria were defined in accordance with a preliminary search strategy, guided by the population, concept and context (PCC) (recommended by the JBI for scoping reviews (21)) and are further described in the published protocol of this study (20)."

Comments from the Reviewer:

What is the reason for "In this study we didn't restrict the period of time or language of the included studies"?

Answer from the authors :

Thank you for your comment, according with your feedback we included a justification why we didn't restrict the period of time in this study (lines 172-174, page 6).

"We did not restrict the period of time or language of the included studies in order to reduce the selection bias and to undertake a comprehensive overview of the existing literature on a topic with still limited number of publications."

Comments from the Reviewer:

At the study selection, did you use specific software or did you do it manually?

Answer from the authors :

Thank you for your comment. We have included the following information (lines 168-169 in the study selection section)

“We used Endnote to import articles from the different databases and we used Rayyan as tool to support the screening of titles and abstracts”.

Comments from the Reviewer:

How many articles are in disagreement during study selection?

Please explain the process in detail to ensure the interrealibility of the review. You may refer to the JBI manual.

Answer from the authors :

Thank you for your comment we have included the suggested information

" A third independent author was involved to solve 4 conflicts in authors' decision, leading to the inclusion of one article . In total, 127 articles were excluded after the screening, and 29 articles ultimately met the eligibility criteria." (lines 228-229, page 7)

We also describe the following statement in the methodology of the study (lines 135-138, page 5) :

“This scoping review is conducted using the six-stage approach proposed by Arksey and O’Malley (26) and is guided by the Preferred Reporting Items for Systematic Reviews and Meta-Analysis Extension for Scoping Reviews to ensure the transparency of the results obtained (27) and follows The Joanna Briggs Institute Methodology for Scoping Reviews (28).”

Reviewer: 3

Dr. Dominic Roche, Cardiff University

Thank you for submitting this paper for peer review. This is an important topic and you have presented some interesting and relevant evidence in your scoping review. Please can I refer you to the following points to consider to and which I hope you find helpful in further develop this manuscript.

Comments from the Reviewer:

1. Is the research question or study objective clearly defined?

No.

You set out ‘objectives’ on page 5 – this section isn’t quite congruent with the main goal and objectives outlined on page 6 ‘identifying the research question’ (for example the statement relating to collecting recommendations from previous studies). I would also like to read more about the interrelated dimensions you refer to in this ‘objectives’ section on page 5 and how this helped to inform the planning, design and execution of your scoping review.

Answer from the authors :

Thank you for your comments. In order to adjust according with your feedback we have made some changes in the page 5 when presenting the objectives of the study . We have described the rational of the study and objetives separately (lines 117-133, page 4 and 5) :

"Study rationale:

In this study we set out to map and frame the main factors that underlie an effective implementation of HCW support interventions. We have defined five interrelated dimensions guided by five main research questions, further described in this study, and we organized them in the acronym OPERA (Figure 1). This structure helped to inform the planning and design of the scoping review, as well as the execution. The defined domains were based on health policy implementation, implementation science principles and Donabedian’s structure–process–outcome quality-of-care model, adapted by Yano (17).”

“Objectives:

We aim to understand what existing structures, organisational factors, relevant actors and operational attributes are present in healthcare support interventions, and to identify what are the recommendations from the included interventions for improving the effectiveness of the support programmes. "

Comments from the Reviewer:

Can you also clarify if the aim of the review is to examine factors / issues relating just to the implementation of the programme / interventions, or are you also interested in outcomes and if so, what are these outcomes beyond implementation? In regard to this, I note that the conclusion presented in your abstract relates to HCWs wellbeing which does suggest you are interested in more than implementation factors."

Answer from the authors :

Dear Dr. Andi Irwan, thank you for your comment. The scoping review is focused on factors / issues related to the implementation of the programme . Based in your feedback we have adjusted some language that might not be clear regarding this issue (e.g "The effectiveness of programme' implementation is highly dependent..." (line 561,page 17), "the set of characteristics and elements necessary for a successful programme' implementation" (lines 553-554, page 17))

The success/effectiveness of these interventions lies in the well-being and occupational health of the healthcare professionals. Although the study's focus is on the implementation factors, inevitably the impact or success of that intervention is also captured.

Comments from the Reviewer:

I would question the use of the term 'key factors' in the main aim – perhaps 'factors' more broadly?

Answer from the authors :

Thank you for your suggestion. We decided to indicate "key" factors since the methodology of the study focused on identifying what are the main factors that underlie an effective implementation of HCW support interventions based on the diagram OPERA. We would really appreciate if the title will be the same as the protocol of the study also published in BMJ Open in order to be related with the previous publication.

Comments from the Reviewer:

Please be consistent with use of 'aims', 'goals' and 'objectives' as these appear to be used interchangeably.

Answer from the authors :

Thank you for your comment. We have adjust the terms according with your feedback.

Comments from the Reviewer:

In relation to the secondary 'objectives' – can you clarify why you are only focusing on interventions relating to when HCWs are physically and / or emotionally affected by PSIs? What is the trigger for the intervention(s) to recognise this has occurred?

Answer from the authors :

Thank you for your questions. We have adjusted the research questions to :

In this study we focus on the main research question :

- What are the key factors that contribute for an effective implementation of interventions to support HCWs after PSIs or other similar stressful situations in health organisations?

To answer the primary research question, five secondary questions were formulated based on the specific objectives and outcomes of interest of the study :

- What are the organisational factors that contribute for an effective implementation of these interventions?
- Who are the relevant actors that contribute for an effective implementation of these interventions?
- What are the contextual factors that contribute for an effective implementation of these interventions?
- What recommendations, as identified in previous studies, can be applied to effectively implement these interventions?
- What are the operational attributes that contribute for an effective implementation of these interventions?

We decided to don't specify the motivation for applying this type of support programmes in the research question. As described in the manuscript , these types of programmes can also be suitable to attend the needs of HCWs that need support after a stressful event not related with a patient safety incident. In this sense, we decided to don't specify the situation that motivate the HCW to attend the programme.

Comments from the Reviewer:

I think more background and contextual information in the introduction would more effectively position this scoping review in the context of the stated aim and objectives. This would include more detail about the second victim phenomenon in the context of patient safety incidents and the potential impact on healthcare workers. I'd also like to read more about the theoretical underpinnings of the interventions / programmes you refer to in your introduction. How have these come about – what is the proposed 'mechanism of action' to support healthcare workers in these situations? For example, you note that there are programmes / interventions designed for decreasing emotional and psychological distress in HCWs – what specifically issues / context are particularly relevant when considering PSIs?

Answer from the authors :

Thank you for your feedback. In order to improve the contextual information focused on the link between patient safety incidents and second victim phenomenon and the potential impact on healthcare workers we have changed the 1st, 2nd , 3rd paragraph and first 2 lines of paragraph 4 of the introduction. We tried to be clear and concise in order to don't overtake to much the word counting of the journal.

Comments from the Reviewer:

You focused on five key domains using the acronym OPERA, guided by the five main questions – are you able to provide more detail here as to how you came to decide on these key domains?

Answer from the authors :

Thank you for your comment we have added this information in the section: study rationale (lines 117-127, page 4 and 5).

Comments from the Reviewer:

I also note on page six that you have included "support interventions in health organisations in which HCWs are physically and/or emotionally affected by PSIs and other distressing situations based on the fact that most support interventions end up being directed toward these types of programmes." I

think this statement needs to be developed in more detail with supporting evidence and perhaps examples of what exactly you mean here and how this relates to the aims and outcomes of your review.

Answer from the authors :

Thank you for your comment, the decision to include “other distressing situations” was made during the review of the study protocol, and we didn’t defined them in order to don’t limit our search to publication that only have considered PSIs.

We have included more detail in the lines 180-184 in the study selection section:

“In this study we focused on identifying the highest number of programmes and other initiatives focused on supporting HCWs after PSI’s. Therefore we included support interventions in health organisations in which HCWs are physically and/or emotionally affected by PSIs and other distressing situations. Based on the fact that there is still a lack of assistance to HCWs to cope in distressing situations, some support interventions are opening their scope of action (9)”.

In the background you also can find some literature that sustain also this information. “A systematic review found that HCWs seek this type of support not only after being involved in a PSI, but also when facing other distressing situations (e.g. emotional distress, torpid evolution of a patient, personal crises, intraoperative mishaps) .” - lines 99-101, page 4

Comments from the Reviewer:

2. Is the abstract accurate, balanced and complete?

Yes. With the caveat that you refer to ‘effective implementation’, whereas your review appears to have addressed issues beyond implementation (or there needs to be more clarity regarding how you have conceptualised implementation in the context of this review). The aim could be separated from the background section so it is more clearly identified.

Answer from the authors :

The abstract was changed according with the editors suggestions to be adapted for review articles:

“Please revise the formatting of your abstract so that it includes the following sections: Objectives >> Design >> Data Sources >> Eligibility Criteria >> Data extraction and synthesis >> Results >> Conclusions.”

Comments from the Reviewer:

6. Are the outcomes clearly defined?

No. As noted in my response to question one, I found it a little challenging to identify what the specific outcomes of interest were. You have stated you are interested in factors related to the implementation of these programmes, but this doesn’t appear to be congruent with some of the objectives stated on page 6 as you then refer to factors contributing to effective support programmes. Do you mean effectiveness of implementation or effectiveness of programme outcomes (or both)?

Answer from the authors :

We highly appreciate your feedback. After your comment we identified that it wasn’t clear in the manuscript our specific outcome of interest.

We have adjusted the manuscript in order to make it clear our main outcomes of interest and that we focused on the implementation process. Namely we have describe them in the objective of the study and on the research questions.

We also have include in the introduction section the following information to make more clear that the success of intervention (in this case to provide timely and adequate support to HCWs after the occurrence of PSIs) it’s highly dependent on an adequate previous planning and implementation

(lines 111-112 – last paragraph of introduction section). We have reviewed all the manuscript to adjust all the potential information that it wasn't clear.

Comments from the Reviewer:

10. Are they presented clearly?

Yes. However, you have used the acronym OPERA to guide your results and provide a helpful figure in support of this. You could also more clearly sequence your results to reflect this acronym, which would help provide a more logical structure to the results section. I would also encourage you to make more effective use of numbering and / or sub-headings in your results section to help guide the reader through the respective sections. (NB on page 7, 'stage 5' you could also present the main categories in the 'OPERA' sequence to provide a more logical structure and clearly identify 'attributes'.)

Answer from the authors :

Thank you so much for your comment. After reading your appreciation we have reorganized the results section in order to respect the order of the acronym OPERA and also we have included some more detailed explanation about the relation with the results and the acronym :

"In the following section we present the results based on the organisation of the acronym OPERA : Organisational factors (structure, process and outcome), People, Environment, Recommendations from the included studies and Attributes of the support programmes." (Lines 254-256, page 8)

We also have made a change in the subheadings of the results section, in order to make the acronym more clear:

- (Organizational factors)PERA;
- O(People)ERA
- OP(Environment)RA
- OPE(Recommendations)A
- OPER(Attributes)

Comments from the Reviewer:

The results are in some cases presented using a list / bullet point approach. I think it would enhance the readability of your results section if you consistently adopted a narrative approach to the presentation by synthesizing the points of discussion together to help more effectively represent the respective themes and sub-themes.

It's unclear to me why you have chosen to present the 'recommendations' theme as a table, whereas the other themes are not presented in this way. (I wonder if this is related to word count limits?) I did find this approach disrupted the flow of your results – are you able to report the recommendations as a narrative, perhaps supported by supplementary information in a table?

Answer from the authors :

Thank you for your comment, since we have made an extensive review with several outcomes (based on the different categories of OPERA framework) it was very challenging to write in a clear and concise way to avoid extending to much the word limit of the journal guidelines. Therefore, we have decided to describe them in bullet points or in short sentences. Taking into account the extensive section of results, we understood that this approach would increase the readability of the article and to make it more friendly to the reader.

Regarding the recommendations theme, we have decided to show the results in a table to also facilitate the readability of the article. We have tried to report the recommendations section in narrative however it will increase a lot the word count of the article. We ask you to take in consideration our feedback and to understand our efforts.

Comments from the Reviewer:

11. Are the discussion and conclusions justified by the results

Yes. However, there are some short paragraphs which disrupt the flow of the discussion section – this is particularly evidence toward the latter stage of the section.

Answer from the authors :

We have reviewed again the discussion section in order to improve the writing and its content.

Comments from the Reviewer:

In regard to the conclusions, I wonder if you would be able to develop a model / framework that can demonstrate to the reader what key factors are required for effective implementation after PSI's in health organisations? This would then clearly address the originally posited question for your review.

Answer from the authors :

Thank you for your comment we have included a summary table where we have organized the main study recommendations according with acronym OPERA after discussing the results of the study (page 16).

Comments from the Reviewer:

I think you could also develop in more detail the issue of non-hospital based interventions – what could be done to address the lack of evidence in this regard?

Answer from the authors :

Thank you for your recommendation we have included some evidence to explain the possible reasons for the implementation types of support programmes still limited in non -hospital settings:

“Although hospital settings are increasingly investing in monitoring programmes, the study of the impact of HCWs’ support programmes is still limited and unreported in non-hospital settings (82). This might be due to the insufficient emphasis on safety culture in non-hospital settings (82,83), which could be improved by increasing patient safety initiatives and awareness campaigns in these contexts.”- (lines 518-521, page 15)

Comments from the Reviewer:

I also think it would be helpful to consider more overtly during your discussion and concluding remarks what we already know about other support mechanisms and interventions for staff that are not specifically related to PSI and the SV phenomenon and what learning we could take from this evidence base.

Answer from the authors :

Thank you for your comments. We have outlined the importance of previous experience from support programmes directly and not directly associated with supporting HCWs after PSIs namely related with identification of potential barriers and challenges experienced by previous programmes, such as confidentiality concerns and professional liability.

Other important notes from the authors: All the references have been reviewed.

VERSION 2 – REVIEW

REVIEWER	Roche, Dominic Cardiff University
REVIEW RETURNED	14-Nov-2023

GENERAL COMMENTS	Thank you for considering my original comments and for sharing the revisions you have made to your manuscript. I have enjoyed reading your very interesting and important review. A minor issue, but there are still a few typos and word choice issues, for example: L75: “between 10.4% and 50%” “the healthcare sector” L82: “in particular” L92: “has a strong...” L109: “focusing” L241: Typo in table 1 – should this be level IV or level VI? However, I am sure these will be corrected with a further proof read.
--

VERSION 2 – AUTHOR RESPONSE

Answer from the authors to Reviewer 3 :

Dear Dr. Dominic Roche,

Thank you for carefully read our manuscript and for the time and thoughtful consideration dedicated to the last review. It was very helpful for improving the manuscript.

We have considered your feedback and made the necessary corrections.

We confirm that the characteristics of the included studies are:

Evidence II (Evidence obtained from Randomised Controlled Trial)

Evidence VI (Evidence from a single descriptive or qualitative study).

We have corrected the typo in line 223 “Studies with levels of evidence II and VI (32) met...”.